# Management of Plant Beneficial Fungal Endophytes to Improve the Performance of Agroecological Practices

**DOI:** 10.3390/jof8101087

**Published:** 2022-10-15

**Authors:** Bouchra Nasslahsen, Yves Prin, Hicham Ferhout, Abdelaziz Smouni, Robin Duponnois

**Affiliations:** 1Laboratoire des Symbioses Tropicales & Méditerranéennes, Institut de Recherche pour le Développement, Centre de Coopération Internationale en Recherche Agronomique pour le Développement, Institut National de Recherche pour L’agriculture, L’alimentation et L’environnement, Institut Agro Montpellier, Université de Montpellier, 34398 Montpellier, France; 2Société Agronutrition, 31390 Carbonne, France; 3Laboratoire de Biotechnologie et Physiologie Végétales, Faculté des Sciences, Université Mohammed V de Rabat, Rabat 10000, Morocco; 4Laboratoire Mixte International—LMI AMIR, Rabat 10000, Morocco

**Keywords:** biostimulant, biofertilizer, fungal endophytes, mycorrhiza helper bacteria, nurse plants, controlled mycorrhization, sustainable agriculture

## Abstract

By dint of the development of agroecological practices and organic farming, stakeholders are becoming more and more aware of the importance of soil life and banning a growing number of pesticide molecules, promoting the use of plant bio-stimulants. To justify and promote the use of microbes in agroecological practices and sustainable agriculture, a number of functions or services often are invoked: (i) soil health, (ii) plant growth promotion, (iii) biocontrol, (iv) nutrient acquiring, (v) soil carbon storage, etc. In this paper, a review and a hierarchical classification of plant fungal partners according to their ecosystemic potential with regard to the available technologies aiming at field uses will be discussed with a particular focus on interactive microbial associations and functions such as Mycorrhiza Helper Bacteria (MHB) and nurse plants.

## 1. Introduction

Biofertilizers are a class of biostimulants for which there is a plethora of definitions: the European Biostimulant Industry Council (EBIC) proposes the following definition: “Plant biostimulants contain substance(s) and/or micro-organisms whose function when applied to plants or the rhizosphere is to stimulate natural processes to enhance/benefit nutrient uptake, nutrient efficiency, tolerance to abiotic stress, and crop quality” [1].

A recent report [2] recalled the abundance of terminology related to plant stimulation products and provided an exhaustive inventory, and we refer the reader to this document. In this review, we will consider microbial biofertilizers or, instead, the microbial component of biofertilizers.

Until the early 1980s, agroecology was considered a desired goal for agricultural systems, aiming at solving the sustainability problem of agriculture. At that time, some transposable field practices were still quite limited, particularly in developed countries where large-scale agrosystems, in order to have a more profitable agriculture, used chemical inputs and an extremely high level of mechanization [3]. Since then, the advent of agro-ecology and organic farming, the awareness of the importance of soil health and the banning of a growing number of pesticide molecules have changed the plant bio-stimulants market. 

Remarkably, a number of scientifical research priority programs and calls, as well as a higher number of private companies and startups in the domains of (i) seed selection, (ii) phytopathology and (iii) chemical fertilizers, are now turning to the acquisition and characterization of microorganisms as alternatives to chemicals fertilizers or biocontrol agents against plant disease or pest attacks, aiming mainly at the improvement of crop production, the interface of partners in crop associations, the shared networking of soils for the rehabilitation of lands or the restoration of different ecosystems. This was made possible by the concomitant emergence of companies developing tools and markers in the (i) global chemistry, (ii) sequencing and (iii) microbiota analyzing domains.

As microbial biofertilizers, both eucaryotes—such as (i) ecto (ECM) and (ii) endotrophic (AM) mycorrhizal fungi—and procaryotes—free or symbiotic (i) nitrogen-fixers, (ii) phosphate-solubilizing bacteria, (iii) plant-growth-promoting rhizobacteria (PGPR), etc.—are considered, and more recently, even viruses are used as bacteriophages for biocontrol purposes. 

Plants can also, sometimes, be used as supports to produce both cultivable and uncultivable microbes, such as Glomeromycetes in endotrophic mycorrhizal symbioses, normally produced on transformed root systems.

More recently, the concept of nurse plants has been developed on the basis of the higher ability of particular plants to mobilize a diversified root microbiote, thus allowing a better soil reactivation and plant growth of their co-cultivated associates through a diversified range of bacterial or mycorrhizal functions. The use of these high potential holobionts was qualified as holistic by several authors [4] (Figure 1).

Challenges to the use of these beneficial microbes are wide and include a number of characters such as their (i) identification among a soil microbial complex or plant microbiota, (ii) the ease and difficulty of their cultivability and (iii) their survival as a conditioning inoculant.

In the acquisition and use of microbial resources, several elements must be considered:-Access to resources: Among crop plant species, domestication and genetic selection often lasted for thousands of years, with more or less long distances of transportation from their native areas to the current zones of production [5]. Native areas should be regarded as a major source of information about the natural microbiota of the considered crop with a set of functions presumably essential to the plant holobiont life cycle naturally developed during evolution and early domestication. Of course, native areas of the considered crop plants must be known, the corresponding countries or zones must be accessible, the ancient ecosystems preserved, etc. [5]. The plant original diversities have to be explored across domestication steps, in several countries, with regard to the more or less progressive (e.g., intra- vs. intercontinental) dissemination. As early as possible in the development of microbial exploration projects, the terms and purposes of use must be established with the partner country or countries and adjusted as the databases evolve and their potential for use is assessed, both for plants and microbial strains (in accordance with ABS rules).-Cultivability or the survival of microbial strains: Having the microbial strain available as a pure culture is a typical objective of an agro-microbiologist; however, non-cultivability is not an indicator of non-viability as evidenced by Xu et al. (1982) on different bacterial taxa [6]. Non-cultivability is considered a general fate of AM fungi (although *Glomeromycota* are still partially explored in terms of cultivability), and ECM are also not always easy to cultivate or to maintain over time in pure cultures. Plate cultures of the ECM ascomycete *Tuber* spp. are possible [7,8], but the presence of bacteria of the genus *Rhodopseudomonas* sp. (Figure 2) seems obligate [9]. Cultivability can thus constitute a real obstacle to agronomic use. Molecular methods of global analyses of microbial communities (metagenomic) can be used in parallel with microbial isolation trials to evaluate the relative rates of non-culturable microbial taxa and thus evaluate the representativity of the isolates.

-Inoculation method: Depending on the plant and the cultivation methods (the need for a nursery stage, direct sowing, mechanized or not, cuttings, grafting, etc.), and also depending on the microbial strain and the form of inoculum, supply must be adapted.

In this paper, we will summarize some of the approaches that can be used to stimulate or reconstitute soil microbial life with regard to the environmental situation in the frame of a collaborative program within a variety of countries. We will also propose in this paper to review and try to hierarchically classify strategies to restore soils ecosystemic potential with regard to the available technologies aiming at field uses. 

## 2. Mycorrhizal Symbioses

As already cited, two main symbioses concern plants and fungi: (i) endotrophic or arbuscular mycorrhizal (AM) and (ii) ectotrophic or ectomycorrhizal (ECM) associations. These microorganisms play a main role in improving plant defense against root pathogens and root browning. They also enhance the performance of their host plant by colonizing and exploiting a much larger volume of the soil than what could be explored by only the root system. The interaction between the host plant and the symbiotic fungus confers multiple advantages for the plant, by allowing a better uptake of water, P, Cu, Zn and nitrogen [10,11,12], by releasing carbonaceous compounds that constitute the mycorrhizosphere [13], by stimulating the production of phytohormones such as abscisic acid [14], by improving plant resistance to (a) biotic stresses [15] and by improving soil structure and stability [10]. Indeed, Tahat et al. [16] insist on the fact that AM fungi improve the functions of the rhizospheric part of the soil and on the beneficial effects they exert on the latter, i.e., on the properties of the soil. Mahmoudi et al. [17] also showed that AM fungi are indicators of soil multifunctionality. Indeed, the multifunctionality of the soil is strongly dependent on the mycorrhizal traits, and the mycorrhizal intensity is more correlated with the multifunctional character of the soil than the mycorrhizal frequency. 

### 2.1. Arbuscular Mycorrhizal Symbioses

AM associations are the most widespread plant symbioses in both natural or cultivated ecosystems as they concern about 74% of terrestrial vascular plants, thus being among the most important symbiosis from an ecological and economic point of view [18]. The fungal partners all belong to the class Glomeromycetes, in which about 250 species have been described in 4 orders [19]. The AM symbiosis is thought to have appeared more than 500 million years ago [20]. During this long period of coevolution, AM fungi became strict symbionts, dependent on their host plant for carbon nutrition [21]. Indeed, many plants of agronomic interest such as rice [22], maize [23] and wheat [24] form AM associations. This symbiosis is characterized by the formation of intracellular fungal organs, arbuscules, vesicles and spores, which are a form of preservation and dispersal of the fungi and are usually in a free form in nature. The fungi sporulation may occur within roots, rhizosphere or soil. It is easily obtained in synthetic medium in the presence of, e.g., transformed carrot roots. As single cell individuals, spores are an important and useful criteria of an AM fungi: beside a classical form of survival in adverse soil conditions, in the absence of a compatible host plant or during winter, they have long been (and remain) a precious tool for first taxonomic identification based on size, color, wall layers, subtending hyphae, etc. Such parameters allow us to physically separate the sporal community of a soil spore extract after, e.g., wet sieving and sucrose gradient centrifugation (see, e.g., [25,26]), into different morphotypes, thus opening the door to a first isolation/purification step. Spores are extracted from soil harvested under the target plant species by standard procedures requiring some lab equipment. These procedures are well detailed in the literature as in Brundrett et al.’s manual [18]. From the raw extract, spores could be separated from each other based on color, size and morphology criteria (Figure 3), and then multiplied, after inoculation, on generalist propagating plants.

The mycorrhizal roots of the propagating plant are then chopped up and mixed with a sterilized medium (peat) for inoculum use [18]. The medium substrate can also be used as inoculum. The conservation of AM fungi strains can be done by regularly transplanting inoculated plant species.

A typical example is *Rhizophagus irregularis*, which is a particularly well-studied AM symbiont that is produced and used in agrosystems. Besides *R. irregularis*, within AM, the number of taxa available as commercial spore inoculums remains relatively limited: a dozen fungal strains can be purchased from GINCO in Canada [27] but only for research purposes and in small amounts (5 to 6 plugs in microtubes), with the need to be able to multiply this inoculum before use in greenhouse or field conditions. Together with GINCO, the center of study on AM monoxenics (CESAMM) at the University of Leuwen in Belgium offers training sessions for researchers and lab technicians to acquire the techniques required to produce AM fungi under *in vitro* conditions, and for research purposes [28]. Some strains may also be acquired together with plant material (e.g., transformed carrot roots), allowing fungal strain cultivation for research purposes. Other AM inoculants may be purchased as ready-to-use inoculant bags containing fungal strains in a substrate as MYC800R containing *Rhizoglomus irregulare* commercialized by Lallemand Plant Care [29]. Pure spore suspension of *R. irregularis* can also be purchased from Agronutrition™ under the commercial name: ConnectisR.

### 2.2. Other Plant-Growth-Promoting Endophytic Fungi Outside the Glomeromycota

Among fungal biofertilizers, two types of fungal associates will be considered hereafter: (i) fungi belonging to the genera *Piriformospora* (now *Serendipita*, even though both terminologies are still being used) and (ii) the group of fungi called “dark-septate endophytes (DSE)” [30].

*Serendipita* (=*Piriformospora*) *indica*: Described in 1998 [31], *Piriformospora indica* was considered an axenically culturable VAM-like fungus and was first placed within *Basidiomycota* (Hymenomycetes), i.e., far from *Glomeromycota* [32,33]. It was isolated from the Thar desert in India [31] and since then has been characterized as a growth-promoting fungus.

After penetrating living plant cells, *P. indica* can live partially or completely inside plant cells. It can be produced on several basic synthetic media (e.g., Potatoe Dextrose Agar, PDA) and induces plant growth promotion on a wide range of terrestrial plants [34]. In 2016, *P. indica* was transferred to the genus *Serendipita* (*S. indica*) within the family *Serendipitaceae*, a sister group from *Sebacinaceae* within the *Sebacinales* [35]. *S. indica* induces plant growth promotion by better nutrient uptake, tolerance to stresses and pathogens, and production of secondary metabolites [36]. Thanks to its cultivability, *S. indica* has been used as a model fungus to analyze the response of legume plants such as alfalfa to combinations of fungal and bacterial inoculants [37]. It can also be studied in association with the non-VAM, model plant *Arabidopsis thaliana* with its abundantly available genetic mutants, to analyze the genetics of the plant response to fungal inoculants [38]. These new taxa clustered among Sebacinales offer a unique opportunity to understand the diversification and evolution among the wide range of fungal associations and symbioses it encompasses.

### 2.3. Dark Septate Endophytic Fungi (DSE)

Since Melin (1923), a number of terms have been inventoried by Jumpponen and Trappe (1998) [39] to qualify diverse pigmented fungi frequently colonizing living plant roots by septate hyphae. They do not form any specialized structure such as arbuscules or hyphal coils within AM fungi-colonized root cells and are generally designated as dark septate endophytic fungi (DSE). DSE are taxonomically placed among Ascomycetes within several orders such as *Helotiales*, *Sordariales*, *Hypocreales*… [40]. The simple observation of pigmented hyphae within a healthy plant root system should question their affiliation to DSE, followed by confirmation with molecular approaches. The absence of a specialized organ within DSE-colonized roots does not facilitate the understanding of the interactions between partners, but several positive plant responses to inoculation with DSE have been evidenced, such as tolerance to environmental stresses (drought, salinity, metallic contaminants) or to pathogens. However, there would be a large range of plant responses to a DSE strain, from parasitism to mutualism, depending on the plant species [40]. Plant growth promotion through the production of molecules such as phytohormones (auxin, gibberellin) [41] or volatile organic compound (VOC) emission [42] have also been reported.

### 2.4. Ectomycorrhizal Symbioses

The second most studied mycorrhizal symbiosis is the ectomycorrhizal (ECM) symbiosis which mainly concerns perennial plants, trees and shrubs. The fungi involved are within Basidiomycetes or Ascomycetes. The number of fungal species is estimated to be between 20,000 and 25,000 [43]. There are three characteristic structural elements of ectomycorrhiza. Firstly, the fungal mantle, visible to the naked eye, which consists of the mycelium of the fungus wrapped around the plant root. Secondly, the extra-matrix hyphae will establish a link between the mantle, the rhizosphere and the soil. They will allow a better exploitation of the soil [21]. Finally, from the mantle, there is the formation of an intercellular mycelial network limited to one or two cell layers of the root (rhizodermal and cortical cells) without penetrating them: the Hartig network, creating a true “host–hyphae–soil” continuum, a place for the exchange of nutrients between the two symbionts. Ectomycorrhizal fungi are able to grow without the presence of the host plant, but the symbiotic association with the latter is necessary to complete their biological cycle. The important role of the fungus under water stress conditions has also been shown [44]. The fungus will increase the supply of minerals with the help of a battery of enzymes that will degrade the organic matter of the soil [45] and thus make available the minerals that were previously inaccessible to the plant, especially P [46]. Ectomycorrhizal symbiosis appeared 130 MA ago and concerns only 3% of vascular plants identified in gymnosperms and angiosperms. ECM symbioses are found in all the major groups of dicotyledons and in one group of monocotyledons (the Cyperales) [18].

Among ectomycorrhizal fungi, some genera are quite easy to isolate or cultivate, and genera such as *Pisolithus*, *Scleroderma*, *Hebeloma* and *Laccaria* are, for example, routinely used as experimental lab models. However, conservation of strains requires regular sub-culturing on agar media which may potentially be time-consuming for large collections. Inoculums are usually produced on a sterilized peat/vermiculite mixture [18,25]. Inoculations can also sometimes be made in the form of crushed sporocarps such as for the production of mycorrhizal plants in nurseries (e.g., truffles).

### 2.5. Mycorrhization Helper Bacteria (MHB)

As reviewed by Lies et al. (2018) [47], mycorrhizal symbioses, be they AM or ECM, develop and function in the presence of other microorganisms, and particularly bacteria, in the rhizosphere, mycorrhizosphere and hyphosphere. These bacteria impact the whole life cycle of the fungus, including spore germination, hyphal growth, nutrient acquisition, symbiotic infection and fructification/sporogenesis. An increasing number of procaryote taxa have been involved in those interactions with different functions. Regarding mycorrhizal symbioses establishment, two scientific articles are often cited: Garbay [48] and Frey-Klett et al. [49]. Since 2007, MHB has generated an impressive number of publications either as original papers or targeted reviews, with the concerned fields of research being extremely wide, from AM to ECM symbioses, including procaryote, glomeromycetes, asco- and basidiomycetes, land plants and the environmental constraints linked to these different associations. The recent approaches of massive sequencing of environmental samples have largely widened the spectrum of potential molecular taxa associated with mycorrhizal symbioses.

Regarding AM symbioses, several types of interactions have been described in the literature and will be examined hereafter: for example, MHB can influence AM establishment as well as spore germination and hyphal growth through the production of phytohormones [50] or of molecules that stimulate root exudate production, activating the hyphal growth and leading to a higher rate of root colonization [50]. Taxonomically, MHBs belong to Firmicutes (mainly *Bacillus* and *Paenibacillus*), Actinobacteria (mainly *Streptomyces*), α-Proteobacteria (mainly *Azospirillum*, *Bradyrhizobium* and *Rhizobium*), β-Proteobacteria (*Burkholderia*) and γ-Proteobacteria (*Azotobacter*, *Klebsiella* and *Pseudomonas*) [47]. In 2018, intimate interactions were described in a Brazilian leguminous tree, *Piptadenia gonoacantha* (Caesalpinioideae), and particularly rhizobial strain-dependency for AM mycorrhization, for nodule effectiveness and plant growth [51]. These same authors also described the co-occurrence of both symbionts within efficient nodules (Figure 4).

In fact, while being hosted by most plants among their microbiome, AM fungi also host their own microbiome in both the hyphosphere and sporosphere community, and in the intrahyphal and intrasporal compartments as endobacteria. Two types of endobacteria have been described so far: *Candidatus Glomeribacter gigasporarum* (a beta-proteobacterium) from the *Gigasporaceae* [52] and a Mollicutes-related bacteria, which is not more taxonomically described [53]. In a molecular survey (using cloning libraries), the occurrence of relatives from both endobacterial taxa was evidenced in the single spores of AM fungi from different geographical origins [54].

A positive impact on mycorrhization may be linked to the stimulation of fungal spore germination or hyphal growth making it possible to have either a faster meeting between the soil growing mycelium and the root system, or a higher volume of soil exploration from the extraradical mycelium. In a recent paper, Emmett et al. [55] observed diversified but conserved bacterial communities associated with the extraradical hyphae of *Glomus versiforme* and *R. irregularis*, without presuming about their beneficial or antagonistic effect in the field. A beneficial effect can also be linked to a better access to soil nutriments due to the enzymatic toolbox of the procaryotes, to a better tolerance of abiotic stresses (salinity, drought, toxicities…), antagonistic effects against pathogens, etc. Signaling between MHB and AM fungus may be with or without contact, e.g., through volatile organic compounds (VOC) [56,57].

Recently, some authors set up a culture medium, IH medium [58], allowing the *in vitro* culture of *R. irregularis* strain DAOM197198. However, the non-cultivability of AM fungi is still a scientifical research challenge [59]. Several bacteria were found to be associated with *Glomeromycota* spores such as *Streptomyces orientalis*, stimulating the germination of *Funneliformis mosseae* [56,57] or *Brevibacillus* sp., stimulating spore production with *Acaulospora tuberculata* [60]. In neo-caledonian conditions, the bacteria *Curtobacterium citreum* stimulated spore density with *Rhizophagus neocalidonicus* and *Claroideoglomus etunicatum* on the *Cyperaceae Tetraria comosa* [61]. The Firmicute *Paenibacillus validus* produces raffinose as a carbon source that accelerates the germination of fertile spores of *R. irregularis* [62]. Several bacterial taxa were reported to be impacting sporulation and hyphal growth such as the genera *Bacillus*, *Paenibacillus*, *Methylobacterium*, etc. They were more precisely described by Lies et al. [47].

## 3. Agroecological Applications

In this paragraph, we are going to illustrate some of the strategies that we used in the past to inoculate crop plants or perennial trees in field trials as presented in Figure 1, from natural or degraded ecosystems (holistic approaches aiming to manage the soil microbiota) by using plants (nurse plants, cover crops and agroforestry practices) or MHB to control mycorrhization with commercial or newly isolated AM spores (reductionist approaches for degraded to highly degraded ecosystems).

### 3.1. Reductionist Approaches—ECM, AM or MHB Field Trials

As stated earlier, not all mycorrhizal taxa, be they AM or ECM, are easy to cultivate, produce and maintain over a long time period. In the same way, some bacteria are qualified as Viable but Non Culturable (VBNC) within genera such as *Rhizobium*, *Pseudomonas* or *Klebsiella* [63]. For AM fungi, Ijdo et al. [64] classified the large-scale production methods inoculants into three types: (i) the solid substrate, (ii) the substrate-free methods using aero- or hydroponics and (iii) the root organ (typically excised root) culture (ROC) methods, in an ascending complexification order. Growth conditions and host plant choice should be in favor of spore production. Several glomeromycetes have been reported as being routinely propagated in such systems, from pure or mixed spore suspension. Generalist mycorrhizal species such as sorghum, onion and *Tagetes* are the most often chosen, but we also use the easily propagated (as cuttings) *Plectranthus* sp. (Labiatae) or *Fragaria* in the lab. Plants are cultivated in pots on a sterilized substrate and may be submitted to a water stress after several growth weeks to enhance sporulation. After a few more weeks, the substrate can be used either as an inoculum for other pot experiments or as a source for spore extraction and purity control of a pure spore inoculum. For ROC production, excised carrot or *Medicago* excised roots can be purchased from GINCO in Belgium or INVAM in USA, as already cited, either pre-inoculated with a collection of AM fungal strain or uninoculated to be used a multiplication system with preliminarily selected and purified spore suspension.

#### 3.1.1. ECM Inoculation of Forest Trees

In Madagascar, leguminous trees such as *Acacia crassicarpa* constitute a big interest for small farmers because they are fast-growing species in local conditions, but also because they are essentially used as fire and multipurpose wood.

With the availability in Madagascar soils of ectomycorrhizal partners that are compatible with the Australia-native *A. crassicarpa* being questionable, a field trial was designed and conducted by Ducousso et al. [65]. Ducousso and his team tested the compatibility of *A. crassicarpa* with two *Pisolithus* spp. strains (441 and COI.007). *Pisolithus* sp., typically easy to grow in lab conditions, and the two inoculums were prepared in Erlenmeyer glass flasks containing 500 mL of an autoclaved (120 °C, 20 min) mixture of vermiculite and peat moss (9:1, *v/v*). The substrate was moistened with 400 mL liquid MNM medium and autoclaved again (120 °C, 20 min). After 6 weeks at 28 °C, the substrate was colonized by the fungal strains. Plants were inoculated 3 weeks after sowing with 50 mL of inoculum deposited at 1 cm under the collar. Control tests were inoculated with sterile inoculant substrate supplemented with 400 mL MNM.

The field trial consisted of three treatments with 36 (6 × 6) trees per treatment. One plot had 36 trees inoculated with *Pisolithus* 441; another had 36 trees inoculated with *Pisolithus* COI 007; the last one had 36 uninoculated trees (control). Plots were separated by one row of *Eucalyptus robusta* with the same tree spacing. At the time they were planted in the field, the height of all the trees was measured, which would be time 0 of the field measurements. The height at ground level was also measured at 6, 13 and 19 months after plantation. 

There was no significant difference between the three plots. The height and circumference at ground level of *A. crassicarpa* at 0, 6, 13 and 19 months after planting were not significantly different, meaning that neither 441 nor COI007 seemed to significantly induce a better growth of the inoculated plants over the control. Strains 441 and COI007 belong to two different *Pisolithus* species and were easily distinguished by molecular approaches, and this was used to identify the strains involved in plant mycorrhization. Sporophores and ectomycorrhizas profiles from the three different plots were not distinguishable and were very similar to the strain 441. These data showed, in field conditions, that such nursery inoculation may be used on a larger scale, to be statistically consolidated, and that *A. crassicarpa* is a highly mycorrhizal tree species, at least with *P. microcarpus* (strain 441). Moreover, the study demonstrates the importance, in field trials, of the tracing of microbial inoculants that will only allow us to reveal a contamination of all the plots, masking the plant-growth-promoting effect over, at least, the uninoculated control. This was already highlighted by Prin et al. [66] in a field inoculation trial of *Acacia mangium* with two strains of *Bradyrhizobium*, which was also in Madagascar.

#### 3.1.2. The Effect of Soil AM Spores Inoculation on Plant Species


The controlled mycorrhization of an exotic tree leguminous, *Acacia holosericea*, in a Sahelian ecosystem (Burkina Faso)


The study published by Bilgo et al. [67] aimed to test in field conditions the impact of an Australian acacia species, *Acacia holosericea*, on the soil nutrient content and microbial life, including the mycorrhizal potential. The impact of planting the exotic *A. holosericea* in the highly degraded sahelian soil on microbial activities was assessed by measuring the patterns of *in situ* catabolic potential (ISCP) of the soil microbial communities [68].

Hence, the impact of controlled AM fungal inoculation of *A. holosericea* seedlings and how it could counterbalance the effects of *A. holosericea* introduction in a correlation with an improved mycorrhizal soil potential was the main aim of this study. The AM inoculum used was *Glomus intraradices* Schenk and Smith (DAOM 181602, Ottawa Agricultural Herbarium). The AM fungi was multiplied on millet (*Pennisetum typhoides* L.) and for 12 weeks under greenhouse conditions using the Terragreen™ substrate. Before inoculation of the Acacia seedlings, the millet plants were uprooted, gently washed and the roots were cut into 0.5 cm long pieces bearing around 150 vesicles cm^−1^. The control treatment was made with non-mycorrhizal millet roots, prepared as above, but without AM inoculation. 

The results of this study showed that: (i) AM inoculation can significantly improve *A. holosericea* growth after 4 years of plantation, and (ii) the introduction of *A. holosericea* trees can significantly modify soil microbial functions.


The impact of mycorrhiza-based ecological engineering strategies on *Ziziphus mauritiana* Lam. and its native mycorrhizal communities on the route of the Great Green Wall (Senegal)


Jujube tree (*Ziziphus mauritiana* Lam.) is a multipurpose fruit tree commonly used in Sahel in West Africa [69], and it has thus been selected by the pan-African Great Green Wall (GGW) project which aims to « green » and fight against poverty, soil degradation and desertification [70]. 

*Z. mauritiana* is highly mycotrophic (AM symbioses), and AM root colonization of seedlings in nursery is essential to plant survival after outplanting in the field [71].

The study by Thioye et al. (2019) [72] evaluated different mycorrhiza-based strategies, i.e., using a *R. irregularis* inoculant—combined or not with rock phosphate fertilizers—on jujube seedling growth, both (i) in nursery and (ii) after outplanting in the field. The ecological impact of each practice has been assessed by New Generation Sequencing (Illumina) monitoring of AM fungal communities one year after plantation. Two *Z. mauritiana* cultivars from different provenances were considered, a local one (Tasset) adapted to the harsh conditions observed on the route of the GGW and an Indian cultivar (Gola) particularly appreciated by West African farmers because of (i) its fruiting precocity ability, (ii) a larger size and (iii) the taste of the fruits. The *R. irregularis* isolate IR27 (syn. *Glomus aggregatum* IR27) [73] provided by the LCM laboratory (IRD, Dakar, Senegal) was used as AM fungal inoculum. It was propagated on maize (*Zea mays* L.) for three months using sterilized sandy soil in a tree nursery.

The study results clearly showed that ecological engineering strategies using *R. irregularis* can significantly promote jujube performance both in the nursery (growth and nutrition) and field (notable rate of survival), with a positive promoting effect on jujube plant height. The comparison between the local jujube plants (Tasset) and exotic jujube cultivars showed a cultivar-dependent response to mycorrhizal inoculation, with a better response of the exotic cultivar in terms of plant-promoting efficiency, a potentially higher persistence of AM fungal inoculum and more limited disturbances of the native AM fungal community. Results demonstrated a major insight to develop and improve the ecological management of jujube orchards on the route of the GGW (Senegal).

#### 3.1.3. MHB (Mycorrhiza Helper Bacteria)

With AM fungi being time-consuming to manipulate and expensive to produce, another alternative could be used to promote the use of MHB taking advantage of the relative omnipresence of AM fungi, as well as their so-called wide spectrum of host compatibility (attested for some well-known taxa). As bacteria from MHB are generally easier to isolate, multiply and produce than AM fungi and some of them are sporulating (such as *Bacillus*, *Paenibacillus*…) and easy to maintain with basic equipment, they could be a convenient way to stimulate AM symbioses at a nursery or even field scale.

Holistic approaches, acting on agricultural or forestry practices: associating grain legumes and cereals or using *Lavandula* as a soil stimulator in association with *Cupressus*

Both field experiments described hereafter were conducted in Morocco not far from Marrakesh. The first one concerned the impact of agronomic practices on wheat/faba bean associations, in relation with soil microbial life. The second one studied the impact of associating *Lavandula* to *Cupressus* in forestry plantations, still in relation to soil microbial life.

#### 3.1.4. Mixed Legume/Cereals Crop Practices

Faba bean (*Vicia faba*) is a Mediterranean leguminous crop that could enable the diversification of agrosystems [74,75], particularly in intercropping and rotation systems for N and P plant nutrition. Wahbi et al. [76] investigated in field conditions the faba bean impacts on soil microbial functionalities with two cereal/leguminous systems (intercropping and rotation) during two growing seasons. The main aims of their study were (i) to evaluate the agronomic performance of wheat/faba bean in intercrop and in rotation and (ii) to monitor the impacts of these cultural practices on the mycorrhizal soil infectivity and on the microbial soil functionalities. Three treatments were conducted, namely durum wheat as a single crop (W) and two cropping systems: (i) durum wheat/faba bean intercrop (WF) and (ii) wheat/faba bean rotation (F + W). The experimental design was based on two factors and four replication blocks. The factors were the cropping system (wheat/faba bean intercropping, rotation or wheat monoculture). The crops were sown in December for two consecutive years using 400 viable seeds per m^−2^ in rows 0.18 m apart for wheat and at a rate of 200 kg·ha^−1^ for faba bean. When intercropped, the two species were sown in the same row to maximize root proximity and plant–plant interactions. After one cropping season, a non-significant effect on the wheat development was recorded between the wheat monoculture and the wheat/faba bean intercropping. After the second cropping season, crop data showed that the total biomass yield, spike number and spike dry weight were higher in the intercropping treatment (WF) compared to the wheat monoculture (W) and faba bean/wheat rotation (F + W) treatments with some enhancements resulting from the intercropping vs. monoculture of +84.2%, +24.8% and 122.7%, respectively. The thousand-seed weight was significantly higher in the WF and W + F treatments compared to the monoculture (W).

In conclusion, the study showed that the benefits of intercropping are highly subjected to the mycorrhizal symbiosis establishment and its impact on the soil microflora functionalities. Over two growing seasons, it was clearly shown that intercropping lead to higher performances in crop yield than the rotation and the monoculture. Benefits resulting from intercropping were highly linked with changes recorded on the mycorrhizal soil infectivity and on soil microbial functionalities.

#### 3.1.5. Nurse Plants in Tree Planting

The purpose of this practice is to associate the capacity of certain plants to mobilize a large diversity of soil microbial partners and to make a crop or a perennial species production take benefit of it. Moreover, the concept of holobiont—associating a host plant from the seed to the fruit with a more or less diversified microbiote, whose composition may be artificially driven [77,78]—is a highly promising avenue of research for the development of agroecological practices. 

*Lavandula*/*Cupressus atlantica* association

In Morrocco, Duponnois et al. (2011) tested the combination of native *Lavandula stoechas*—used as “nurse” plants—on the growth of *Cupressus atlantica*, on microbial activity and on arbuscular mycorrhizal (AM) soil potential [79]. An experimental field plantation was designed to compare the AM inoculation *C. atlantica* with native AM fungi and the combination of (coplanting) *C. atlantica* with local *L. stoechas*. The control tests were either without AM inoculation or without *L. stoechas*. The native spore inoculum was prepared from soil samples collected in *C. atlantica* rhizosphere. The spores were extracted by wet sieving on a sucrose gradient and grouped based on their morphological characteristics under a stereomicroscope. They were propagated on maize (*Zea mays* L.) according to Ouahmane et al. (2007) [80,81]. Three years after plantation, this association between *C. atlantica* and *L. stoechas* lead to a higher growth of *C. atlantica* and better soil microbial characteristics compared to the control treatment. AM mycelium network, total microbial activity, dehydrogenase activity, phosphate-solubilizing fluorescent pseudomonads and N, P nutrient uptake by *C. atlantica* were significantly higher in the presence of *L. stoechas* than those recorded in the other treatments. This pioneer shrub thus facilitated the early establishment of Cypress seedlings by improving soil microbial characteristics and AM fungus community development. Given that the facilitative effect of one plant species to other increases with abiotic stress, the benefit of this technique seems very promising in reforestation programs aiming to rehabilitate degraded areas in the Mediterranean region.

## 4. Conclusions

According to several scientifical studies, reviews and calls, the agronomic potential of endophytic root fungi is attested. However, despite their major importance in soil health management, their large-scale use is limited due to various factors such as constraints related to the production of fungal inoculants or the variability of their effects on target plants. The management of the mycorrhizal soil infectivity should be addressed considering the concepts of soil microbial ecology, and the valorization of these biofertilizers should not follow the usual rules of standard fertilization. Studies must be developed to evaluate the different strategies to be implemented (holistic or reductionist approaches) to sustainably manage the biological and chemical fertility of agrosystem soils.

## Figures and Tables

**Figure 1 jof-08-01087-f001:**
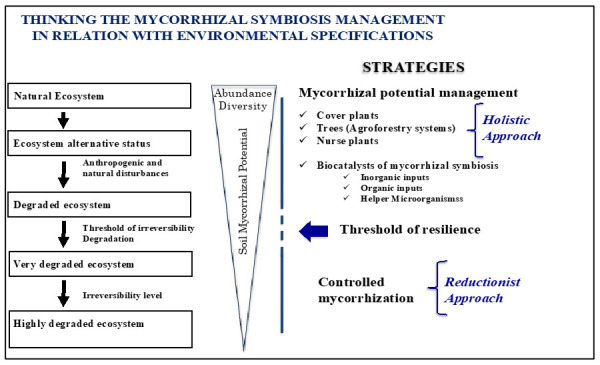
Strategies for managing the mycorrhizal infection potential (MIP) according to the extent of degradation (resilience threshold) of the environment to be remediated. Holistic approach: increased MIP via biological vectors (cover plants, nursery plants, etc.). Reductionist approach: mass introduction of mycorrhizal or MHB inoculant into the environment to be remediated (controlled mycorrhization technique).

**Figure 2 jof-08-01087-f002:**
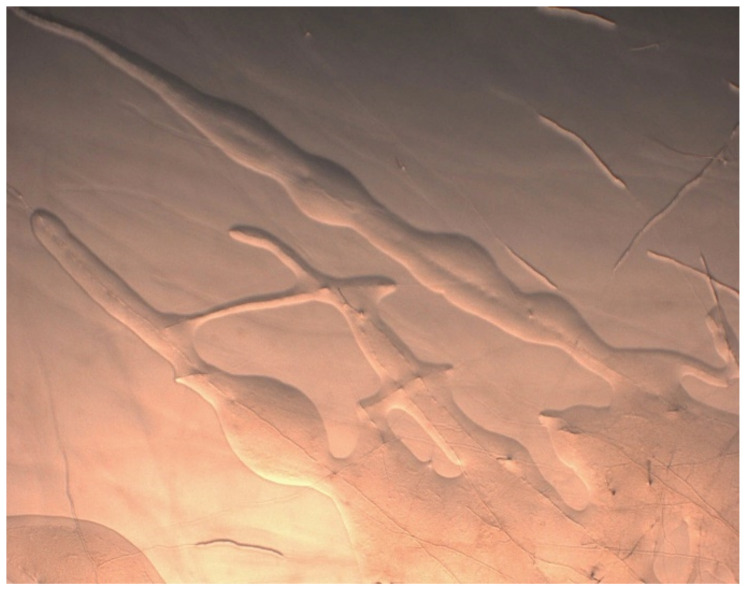
Growing hyphae at the periphery of a Tuber melanosporum colony, ensheathed in *Rhodopseudomonas* sp. colonies [9].

**Figure 3 jof-08-01087-f003:**
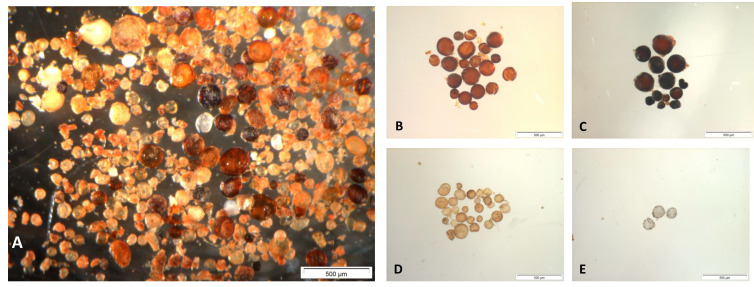
(**A**) Crude extract of spores of *Glomerycota* from a soil under argan tree in Morrocco. Extraction procedure using a saccharose gradient according to Brundrett et al. [18]. (**B**–**E**) Some of the spores of the crude extract obtained after sorting under a stereomicroscope.

**Figure 4 jof-08-01087-f004:**
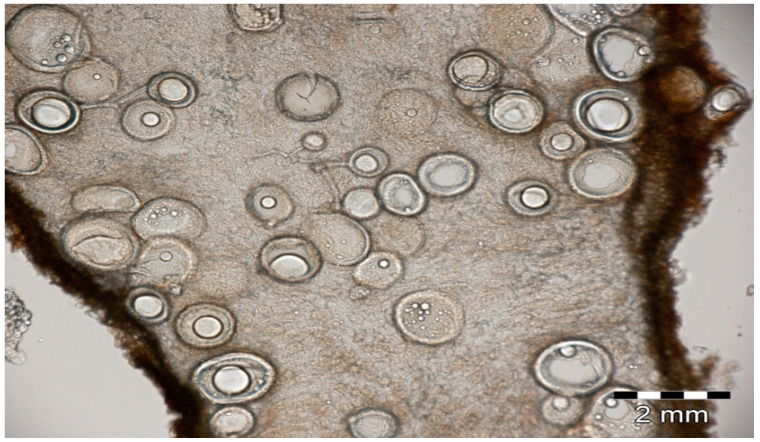
Longitudinal section of a *Piptadenia* sp. nitrogen-fixing nodule with numerous spores and hyphae [51].

## Data Availability

Not applicable.

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
