# Peer review of "Management of Plant Beneficial Fungal Endophytes to Improve the Performance of Agroecological Practices"

_jof, 2022, doi:10.3390/jof8101087_

Round 1

Reviewer 1 Report

Bouchra NASSLAHSEN and coworkers reports a comprehensive information about plant beneficial fungal endophytes the performance of agroecological practices. This goal was achieved through the analysis of available literature, and the review discussions identification, cultivability and availability of microbial. That's a good Angle.

This paper makes a good summary of the previous related research content. Despite these premises, the article introduced the definition of plant stimulating products, but for this article, it is more described as microbial fertilizers. It is worth thinking whether the use of microbial fertilizers is more suitable for the content of the article. As a review, it is worth considering whether an overall system description of the application prospect of plant fungal endophytes in agricultural systems is needed. The use of punctuation in this article still needs attention, such as line 23 and line 73 “:”. There is a lack of references to other articles in the preface, for example, lines 40-56 do not contain a reference.

Author Response

Bouchra NASSLAHSEN and coworkers reports a comprehensive information about plant beneficial fungal endophytes the performance of agroecological practices. This goal was achieved through the analysis of available literature, and the review discussions identification, cultivability and availability of microbial. That's a good Angle.

- Thanks for your appreciation

This paper makes a good summary of the previous related research content. Despite these premises, the article introduced the definition of plant stimulating products, but for this article, it is more described as microbial fertilizers. It is worth thinking whether the use of microbial fertilizers is more suitable for the content of the article. As a review, it is worth considering whether an overall system description of the application prospect of plant fungal endophytes in agricultural systems is needed.

We agree with that and we know that we are far from a consensual definition of the terminology about biostimulants, which is apparently the most inclusive term for a large class of products defined either on their composition, their impact  either on plant on soil on a microbial community on global soil activity on plant growth promotion, on soil microbial diversity, on ecosystems services, with a notion of ecosystems that goes from mixes of plants (natural, planted,...), to the plant itself, its root system  (rhizoplane, rhizosphere, endosphere) from the simplest that comprises e.g. a single bacterial strain within a simple encapsulation formulation (e.g. alginate, peat,..) to more complex microbial mixes (eg ECM, AM, MHB) to plant themselves, pre-inoculated or not, used as microbial support or microbial diversities stimulants. It is clearly one of the aims of this paper to give the reader an image of this widen diversity of products and scales, with their more or less well-defined associated terminology.

The use of punctuation in this article still needs attention, such as line 23 and line 73 “:”. There is a lack of references to other articles in the preface, for example, lines 40-56 do not contain a reference.

We had a careful review of our text and we hope that the numerous corrections and rephrasing we made are now acceptable.

Best regards

Reviewer 2 Report

In this review, a classification of plant fungal partners together with their effects on plant growth and health is presented. However the paper is much more focused on mycorrhizal than in non mycorrhizal endophytic fungi, despite these latter are widely recognised as essential for plant growth promoting and for plant defense from biotic/abiotic stresses. To my opinion at least two paragraphs should be spent to extensively describe these fungi and their applications. In fact par. 2.1.1 and 2.1.2 are about endophytic fungi but in only Piriformospora (Serendipita) species are discussed and DSE species are poorly described. Moreover these paragraph are erroneously put under the main par. 2 Mycorrhizal symbioses.

Another major concern is about Agroecological applications. In this part it seems to me that only applications concerning an improved plant growth are presented, whereas no applications concerning plant defense against biotic/abiotic stresses are discussed.

English should be extensively checked

Other considerations:

L15: Why "hierarchical classification"? Is there a hierarchy in the ability of different groups of fungi in providing beneficial effects to plants?

L27: the tolerance is also to biotic stresses

L54-56: please explain in more detail and add a reference for these sentences. What is the relevance of such fungi?

L57-60: reword this sentence and add references

L69-70 Put "their" after the number

L73, 86 and 101: put anumbered list

L110-120: some sentences are repetitive, english must be corrected. What do you mean for root browsing?

L120-123: reword the sentence

L123: please explain in what sense AM fungi are indicators of soil multifunctionality

L141: ....useful criteria for the identification of AM fungi....

L150-152

Author Response

Dear colleague,

We thank you for their comments and suggestions, and have replied to you as follows

In this review, a classification of plant fungal partners together with their effects on plant growth and health is presented. However, the paper is much more focused on mycorrhizal than in non-mycorrhizal endophytic fungi, despite these latter are widely recognised as essential for plant growth promoting and for plant defense from biotic/abiotic stresses. To my opinion at least two paragraphs should be spent to extensively describe these fungi and their applications. In fact, par. 2.1.1 and 2.1.2 are about endophytic fungi but in only Piriformospora (Serendipita) species are discussed and DSE species are poorly described. Moreover, these paragraphs are erroneously put under the main par. 2 Mycorrhizal symbioses.

We understand the interest of Piriformospora (Serendipita) and DSE species and are convinced of their interest for the understanding of plant-fungi interactions in general. However, we have little practical (field) experience with these classes of fungal plant associates, and our aim in this paper was to give a return of our experiences of field practices and experiments we had in the past, linked to symbioses sensu lato. We mentioned both these fungal associates in the part on mycorrhizal symbioses but clearly mentioned that they were apart "...Among fungal biofertilizers, and despite they should not be considered as mycorrhizal symbionts [19], two types of fungal associates will be mentioned hereafter:.."

Another major concern is about Agroecological applications. In this part it seems to me that only applications concerning an improved plant growth are presented, whereas no applications concerning plant defense against biotic/abiotic stresses are discussed.

We agree that the final aim of our field experiments, as presented in Fig S1 is the plant growth promotion and soil microbial life stimulation but our microbial life analyses (PIM, respirometry, 15N natural abundance, P solubilization...)  often illustrate the concomitant activation of microbial functions in the soil or rhizosphere. Our experiences with plant defense against biotic/abiotic stresses were often realized in greenhouse or nurseries and not in field conditions.

English should be extensively checked

See above our response to reviewer 1

Other considerations:

L15: Why "hierarchical classification"? Is there a hierarchy in the ability of different groups of fungi in providing beneficial effects to plants?

No, hierarchy here refers to the level of ecosystem degradation (Fig 1 and S1) and the corresponding proposed strategies of restoration

L27: the tolerance is also to biotic stresses

We checked again, and this is the definition given by the European Biostimulant Industry Council (EBIC)

L54-56: please explain in more detail and add a reference for these sentences. What is the relevance of such fungi?

This sentence has been deleted as the information was not relevant

L57-60: reword this sentence and add references

This sentencePlants can also, sometimes, be used as supports to produce both cultivable and uncultivable microbes, like are Glomeromycetes in endotrophic mycorrhizal symbioses, classically produced on transformed root systems” has been changed by “Plants can also, sometimes, be used as supports to produce uncultivable microbes, like are Glomeromycetes in endotrophic mycorrhizal symbioses, classically produced on trans-formed root systems  [28] S. Declerck, H. D. de Boulois, M. de Stage, and M. Chave, “Evaluation in vitro du potentiel bioprotecteur des cham-pignons mycorhiziens à arbuscules contre le flétrissement bactérien de la tomate,” p. 69.

L69-70 Put "their" after the number

It has been done

L73, 86 and 101: put anumbered list

It has been done

L110-120: some sentences are repetitive, english must be corrected.

We tried to reduce the repetitive sentences. See also above in our response to reviewer 1

 What do you mean for root browsing?

Sorry, we meant "root browning": corrected

L120-123: reword the sentence

The sentence has been reworded as follow: The production of phytohormones such as abscisic acid is stimulated [14]. Plant resistance to (a) biotic stresses [15] as well as soil structure and stability are improved [10]. Indeed, Tahat et al. (2020) [16] highlighted the fact that AM fungi improve the functions of the rhizospheric soil and on the beneficial effects they exert on the soil characteristics.

L123: please explain in what sense AM fungi are indicators of soil multifunctionality

The sentence has been modified as follow: Mahmoudi et al. (2021) [17] also showed that AM fungi play a major role in terrestrial ecosystems functioning. Indeed, the multifunctionality of the soil (ecosystem soil func-tions) is strongly dependent on the mycorrhizal traits and the mycorrhizal intensity is more correlated with the multifunctional character of the soil than the mycorrhizal fre-quency. 

L141: ....useful criteria for the identification of AM fungi....

It has been corrected

Best regards

Round 2

Reviewer 2 Report

In their reply the authors state “our aim in this paper was to give a return of our experiences of field practices and experiments we had in the past, linked to symbioses sensu lato. “ Generally review papers include not only the personal experiences of the authors. In this sense I suggest to broaden the discussion about endophytic fungi by providing further examples of useful biofertilizers.

The reply to my comment L54-56 is put under comment L57-60 and no reply is provided for comment L57-60

L19: Helper

L60: the reference number should be [4] according to previous numbers

L73: (iii) their…

Fig. 2 caption: use italics for species names here and throughout the ms. Use the numbering for the reference.

L119: root browning

L120-121: “by colonizing…..roots alone” must be reworded

L121: “the plant species considered”. Did you consider certain plant species?

L123: plant nutrition was already mentioned above (L118), please avoid repetitions

L124: the release of C-compounds in the mycorrhizosphere is not properly an advantage for the plant.

L125: ref. No. [14] seems about phytohormone production by Rhizobacteria, please check if it is proper in this part about mycorrhizal symbioses

L127: delete years from references here and elsewhere

L154: target

L155: literature

L157: numbering not needed

I recommend to move 2.1.1 and 2.1.2 in a separate section from Mycorrhizal symbioses

L255: “are often easy”. Tuber spp. For example are not so easy.

L262: Mycorrhiza Helper Bacteria

L268: have been found to be involved

L298: delete “not more taxonomically described”

L313: set up

L330: control

L340: Several

L342-343: delete sentence

L343: Generally. Tagetes?

Check reference numbering, complete citations, some are with et al.

Author Response

Dear Reviewer,

We thank you for your comments and suggestions, and have replied to you as follows:

In their reply the authors state “our aim in this paper was to give a return of our experiences of field practices and experiments we had in the past, linked to symbioses sensu lato. “Generally, review papers include not only the personal experiences of the authors. In this sense I suggest to broaden the discussion about endophytic fungi by providing further examples of useful biofertilizers.

- We already published a number of papers as reviews or more or less "opinion" papers about land degradation and restoration, uses of mycorrhizal or bacterial fertilizers (e.g. Duponnois et al., 2013; Lies et al., 2018; Lhoir et al, 2021) and our aim in this paper was to focus on our own experience with field inoculation trials, to illustrate the diversity of practices that can be used depending on the ecosystems, the environmental conditions, the type of plant and microbes, the type of forest or crop productions... It was not our aim to relate what has been reported by other authors in an extensive review of field inoculation articles that would completely change the objectives, organization and logic of the ms and make it considerably longer.

The reply to my comment L54-56 is put under comment L57-60 and no reply is provided for comment L57-60: we rephrased this part with regards to Doebley et al (5) review on crop domestication.

L19: Helper: done

L60: the reference number should be [4] according to previous numbers: done

L73: (iii) their… done

Fig. 2 caption: use italics for species names here and throughout the ms. We submit our ms with Latin names in italics. It is through JOF editing process that italics disappeared but may be were not lost by the editing service?

Use the numbering for the reference: done.

L119: root browning: done

L120-121: “by colonizing…..roots alone” must be reworded: done

L121: “the plant species considered”. Did you consider certain plant species? : done

L123: plant nutrition was already mentioned above (L118), please avoid repetitions: done

L124: the release of C-compounds in the mycorrhizosphere is not properly an advantage for the plant: we believe that the release of sugars that will constitute the mycorrhizophere is important by allowing the colonization by bacteria with procaryotic functions like nitrogen fixation, P solubilizing activities, ...

L125: ref. No. [14] seems about phytohormone production by Rhizobacteria, please check if it is proper in this part about mycorrhizal symbioses The paper by Cassan et al (14) contains exhaustive information about the phytohormones auxins, gibberellins, cytokinin,... and thus seems well appropriate in the context of phytohormones production and plant mycorrhizae interactions.

L127: delete years from references here and elsewhere: done

L154: target: done

L155: literature: done

L157: numbering not needed: done

I recommend to move 2.1.1 and 2.1.2 in a separate section from Mycorrhizal symbioses: done (2.2 and 2.3)

L255: “are often easy”. Tuber spp. For example are not so easy: this has been rephrased

L262: Mycorrhiza Helper Bacteria

L268: have been found to be involved

L298: delete “not more taxonomically described”

L313: set up

L330: control: done

L340: Several: done

L342-343: delete sentence:  this has been rephrased as well as in paragraph 2.1

L343: Generally. Tagetes?: done

Check reference numbering, complete citations, some are with et al.: done

We hope that all the corrections made will satisfy both reviewers

Best regards
